# Tracking Response and Resistance in Acute Myeloid Leukemia through Single-Cell DNA Sequencing Helps Uncover New Therapeutic Targets

**DOI:** 10.3390/ijms251810002

**Published:** 2024-09-17

**Authors:** Samantha Bruno, Enrica Borsi, Agnese Patuelli, Lorenza Bandini, Manuela Mancini, Dorian Forte, Jacopo Nanni, Martina Barone, Alessandra Grassi, Gianluca Cristiano, Claudia Venturi, Valentina Robustelli, Giulia Atzeni, Cristina Mosca, Sara De Santis, Cecilia Monaldi, Andrea Poletti, Carolina Terragna, Antonio Curti, Michele Cavo, Simona Soverini, Emanuela Ottaviani

**Affiliations:** 1Department of Medical and Surgical Science (DIMEC), University of Bologna, 40138 Bologna, Italy; agnese.patuelli2@unibo.it (A.P.); lorenza.bandini92@gmail.it (L.B.); dorian.forte2@unibo.it (D.F.); jacopo.nanni5@unibo.it (J.N.); alessandra.grassi11@unibo.it (A.G.); gianluca.cristiano3@unibo.it (G.C.); claudia.venturi@aosp.bo.it (C.V.); giulia.atzeni4@unibo.it (G.A.); cristina.mosca5@unibo.it (C.M.); sara.desantis9@unibo.it (S.D.S.); cecilia.monaldi2@unibo.it (C.M.); andrea.poletti9@unibo.it (A.P.); michele.cavo@unibo.it (M.C.); simona.soverini@unibo.it (S.S.); 2Istituto di Ricovero e Cura a Carattere Scientifico (IRCCS) Azienda Ospedaliero, Universitaria di Bologna, Istituto di Ematologia “Seràgnoli”, 940138 Bologna, Italy; enrica.borsi2@unibo.it (E.B.); mancini.manuela@aosp.bo.it (M.M.); martina.barone5@unibo.it (M.B.); valentina.robustelli@aosp.bo.it (V.R.); carolina.terragna@aosp.bo.it (C.T.); antonio.curti2@unibo.it (A.C.); emanuela.ottaviani@aosp.bo.it (E.O.)

**Keywords:** acute myeloid leukemia, FLT3 mutations, tyrosine kinase inhibitors, clonal architecture, clonal evolution

## Abstract

Acute myeloid leukemia (AML) is an aggressive hematologic neoplasia with a complex polyclonal architecture. Among driver lesions, those involving the *FLT3* gene represent the most frequent mutations identified at diagnosis. The development of tyrosine kinase inhibitors (TKIs) has improved the clinical outcomes of *FLT3*-mutated patients (Pt). However, overcoming resistance to these drugs remains a challenge. To unravel the molecular mechanisms underlying therapy resistance and clonal selection, we conducted a longitudinal analysis using a single-cell DNA sequencing approach (MissionBioTapestri® platform, San Francisco, CA, USA) in two patients with FLT3-mutated AML. To this end, samples were collected at the time of diagnosis, during TKI therapy, and at relapse or complete remission. For Pt #1, disease resistance was associated with clonal expansion of minor clones, and 2nd line TKI therapy with gilteritinib provided a proliferative advantage to the clones carrying *NRAS* and *KIT* mutations, thereby responsible for relapse. In Pt #2, clonal architecture was less complex, and 1st line TKI therapy with midostaurin was able to eradicate the leukemic clones. Our results corroborate previous findings about clonal selection driven by TKIs, highlighting the importance of a deeper characterization of individual clonal architectures for choosing the best treatment plan for personalized approaches aimed at optimizing outcomes.

## 1. Introduction

Acute myeloid leukemia (AML) is a clonal hematopoietic malignancy driven by a wide range of genomic alterations that impair the differentiation capacity of hematopoietic stem cells (HSCs) [1,2]. The FMS-like tyrosine kinase 3 (*FLT3*) gene is frequently altered in AML, is a crucial player in leukemogenesis, and confers a dismal prognosis. Physiologically, FLT3 is expressed by hematopoietic stem and progenitor cells and is activated by the binding of the FL ligand [3]. Activated FLT3 leads to cell survival, proliferation, and differentiation of HSCs through the induction of downstream signaling pathways, including *PI3K*, *RAS*, and *STAT5* [4,5,6,7]. *FLT3* mutations are detected in about 30% of newly diagnosed AML cases patients (pts) and include either internal tandem duplications (ITDs, in approximately 25% of patients) or point mutations in the tyrosine kinase domain (TKD, 7–10% of patients) [8,9]. Both *FLT3-ITD* and *FLT3-TKD* result in sustained activation of FLT3 kinase activity and its downstream signaling pathways, fostering the proliferation and survival of AML cells [4,7,10,11,12]. The introduction of small-molecule tyrosine kinase inhibitors (TKIs) designed to target the FLT3 tyrosine kinase receptor has significantly improved both overall survival (OS) and event-free survival (EFS) rates in patients with *FLT3*-mutated AML [9]. Notably, midostaurin has been the first inhibitor approved as front-line therapy in combination with intensive chemotherapy, showing high response rates with prolonged OS [13,14], and has become the current standard-of-care therapy to treat fit, newly diagnosed *FLT3*-mutated AML patients. Second-generation FLT3 inhibitors, characterized by increased potency and selectivity, have been developed in different settings, including relapsed/refractory (R/R) diseases [10,15]. Among them, gilteritinib showed a significant benefit in terms of OS in comparison to salvage chemotherapy in R/R AML patients and received approval in this patient’s setting [16].

Despite the initially high response rates, the resistance mechanisms responsible for relapse have been identified [17]. The failure of TKI therapies has mainly been linked to two different mechanisms. The first type is represented by *on-target* resistance mutations, such as point mutations altering key residues of the FLT3 protein, such as D835, F691, Y842, and N676 [7,8,9,10]. The second type is represented by *off-target* resistance mechanisms, which are FLT3-independent and involve activation of the PI3K/mTOR and JAK/STAT5 signaling pathways [11], as well as mutations in the RAS/MAPK pathway genes, such as *NRAS*, *KRAS*, *PTPN11*, *CBL*, and *BRAF* [9]. Specifically, the adoption of novel and potent TKIs has been shown to exert a selective pressure responsible for clonal evolution, resulting in the acquisition of mutations in founder clones or in the selection of minor resistant subclones [18,19,20]. Our study aimed to take advantage of single-cell DNA sequencing (scDNAseq) to deeply dissect the clonal evolution of two patients with *FLT3*-mutated AML undergoing FLT3 inhibitor treatment. To gain novel insights into how therapeutic interventions shape the clonal hierarchy and drive the evolutionary dynamics of *FLT3*-mutated AML, we conducted a longitudinal analysis of samples collected at the time of diagnosis, during TKI therapy, and at either disease progression (patient 1) or complete remission (CR) (patient 2). 

## 2. Results

### 2.1. Clinical, Cytogenetic and Molecular Patients’ History

In this study, we present the cases of two patients who were diagnosed with AML and treated at the UO Hematology “Lorenzo e Ariosto Seràgnoli” in Bologna. Patient 1 (henceforth referred to as Pt #1) was an 82-year-old man with AML secondary to myelodysplastic syndrome (MDS). He was initially diagnosed with MDS with excess blasts and classified as MDS-excess blasts 2 according to the WHO 5th edition [21], with trisomy 11 detectable by cytogenetic analysis (R-IPSS risk score high). Next-generation sequencing (NGS) analysis using the Myeloid Solution panel (SOPHiA Genetics) identified several pathogenetic somatic mutations: *ASXL1^E646Wfs*12^* (variant allele frequency (VAF) = 22%), *SRSF2^P95H^* (VAF = 49%), *TET2^E843*^* and *TET2^N191Kfs*4^* (VAF = 49% and 46%, respectively), and *RUNX1^R201Q^* and *RUNX1^S322Nfs*1^* (VAF = 20% and 24%, respectively) (M-IPSS risk score high). The patient received standard-of-care therapy with Azacitidine, but after 17 months, he developed hyperleukocytosis (white blood cell (WBC) count, 111.000/mm^3^), with a consequent disease assessment documenting progression to AML. Specifically, bone marrow (BM) evaluation revealed 50% of blast cells at cytological evaluation, expressing CD34+, CD33+, CD117+, HLA-DR+, and MPO+, along with partial expression of CD4, CD7, and TdT by flow cytometry. Molecular analysis revealed the acquisition of an *FLT3^D835H^* mutation, and chromosome banding analysis showed a normal karyotype (NK). Thus, the final diagnosis was AML with myelodysplasia-related gene mutations at adverse risk, according to both ELN17 and ELN22 risk stratifications [9,22]. Azacitidine, in association with venetoclax, was administered as first-line AML therapy. BM assessment after two cycles of this therapeutic regimen showed R/R disease. Due to the presence of the *FLT3^D835H^*, salvage therapy with gilteritinib was started, initially at the standard dose (120 mg daily dose), then increased up to 200 mg/d. After a transient partial response, a subsequent disease assessment after 3 months of gilteritinib therapy revealed disease progression, with *FLT3^D835H^* no longer detectable by restriction enzyme cleavage and gel electrophoresis. Pt #1 had an OS of about 2 years from MDS and 5 months from AML diagnosis.

Patient 2 (Pt #2) was a 50-years-old woman diagnosed with de novo AML due to 50% of myeloid blast cells documented at diagnostic BM evaluation, expression of CD34+, CD117+, CD33+, CD13+, HLA-DR+, CD4+, and CD25+ immunophenotypic markers, NK by chromosome banding analysis, an *FLT3-ITD* (allelic ratio, AR = 0.8), and an *NPM1* mutation (*NPM1*^mut^/ABL × 100 = 262.16 by real-time RT-PCR) at conventional molecular analysis. Thus, the final diagnosis was AML with mutated *NPM1*, with an intermediate risk according to the ELN2022 risk classification [9,22]. She received standard induction therapy with midostaurin in association with intensive chemotherapy (3 + 7 schedules) and achieved complete remission (CR). *FLT3-ITD* became undetectable by fragment analysis, but *NPM1* was still detectable by RT-PCR (*NPM1*^mut^/ABL × 100 = 7.6) and by flow cytometry-based measurable residual disease (MRD, 0.18%). She then received two consolidation cycles of high-dose cytarabine in combination with midostaurin, with persistent morphologic CR but *NPM1* positivity. The patient ultimately underwent an allogeneic BM transplant from an HLA-matched sibling donor and is currently alive and in good clinical condition 42 months after AML diagnosis.

Table 1 summarizes baseline patients’ characteristics. Figure 1 depicts changes in BM blasts (%) during treatment, with a special focus on key time points (response/resistance; therapeutic modifications).

### 2.2. Clonal Architecture and Evolution of FLT3-Mutated AML

High-throughput scDNAseq using the Tapestri platform was performed on mononuclear cells isolated from Pt #1 at different time points: at the time of initial MDS diagnosis, at the time of disease evolution to AML, upon gilteritinib treatment, and during further disease progression. Similarly, for Pt #2, sequencing was performed at the time of AML diagnosis and at two key follow-up time points (f-up): post-induction (after a combination of standard chemotherapy, 3 + 7 cycles, with midostaurin) and post-consolidation therapy with high-dose cytarabine in combination with midostaurin.

For Pt #1, 7188 cells were sequenced across four time points (353, 931, 671, and 5233 for each time point, respectively). Analysis of the clonal architecture of Pt #1 showed a progressive increase in mutation number and clonal complexity during evolution from MDS to AML, and further during AML progression under TKI selective pressure. The highest number of clones was observed in the two f-up samples. Specifically, we detected three mutated clones at MDS diagnosis, seven mutated clones at AML diagnosis, and 10 mutated clones at f-up (Figure 2A). At the time of MDS, the identified clonal populations were characterized by alterations in epigenetic modifiers, with a dominant single-mutated clone carrying *TET2^N191Kfs*4^* (referred to as C1) and two minor related clones that differed in zygosity, with double-mutated *TET2^N191Kfs*4^/RUNX1^R201Q^_Heterozygous* (HET) and *TET2^N191Kfs*4^/RUNX1^R201Q^_Homozygous* (HOM) populations (referred to as C2 and C3, respectively). AML onset was characterized by the branching evolution of two minor clones with the acquisition of pathogenic mutations in *FLT3*, *NRAS*, and *KIT* signaling genes. We detected two dominant subclones composed of *TET2^N191Kfs*4^/RUNX1^R201Q^_HOM/FLT3^D835H^* (C4) and *TET2^N191Kfs*4^/RUNX1^R201Q^_HET/FLT3^D835H^* (C5), along with two minor subclones *TET2^N191Kfs*4^/RUNX1^R201Q^_HOM/NRAS^G12A^* (C6) and *TET2^N191Kfs*4^/RUNX1^R201Q^_HOM/KIT^D816H^* (C7). Lastly, the two f-up samples on treatment and at disease progression showed the emergence of three subclones evolved from C2 and C3: *TET2^N191Kfs*4^/RUNX1^R201Q^_HOM/KIT^D816V^* (C8), *TET2^N191Kfs*4^/RUNX1^R201Q^_HET/KIT^D816H^* (i.e. C9), *TET2^N191Kfs*4^/RUNX1^R201Q^_HET/NRAS^G12A^* (C10). A detailed description of the clonal and sub-clonal populations, together with their relative abundances, is presented in Table 2 and Appendix A.

Next, we explored the clonal repertoire, with clones defined as cells with identical mutational status, and their evolution at different time points. Figure 2B illustrates the distribution of all clonal populations with a given genotype identified by scDNAseq. Pt #1 had a complex genetic architecture, with evidence of ten different clones (C1 to C10, as summarized in Figure 2B), with the founder clone (C1) harboring a *TET2* frameshift mutation (*TET2^N191Kfs*4^*). C2 and C3 evolved from C1 during disease progression by acquiring new pathogenic mutations in *RUNX1*, differing from each other in the zygosity of such mutations. Notably, clone 3 warranted particular attention, as the Tapestri Insight version 2.2 software (Mission Bio, Inc., San Francisco, CA, USA) identified it as harboring a *RUNX1^R201Q^* homozygous mutation. This raised the possibility of being an allele dropout (ADO), i.e., a failure to amplify the wild-type (*WT*) allele at chr21:36231782. Alternatively, it could be the result of a genuine event, such as copy number variation (CNV) or copy neutral loss of heterozygosity (LOH). To confirm or reject CNV interpretation, we performed an analysis using the Mosaic package version 2.3 (Mission Bio, Inc., San Francisco, CA, USA) by focusing on two variants (“chr21:36231782:C/T”, “chr4:106155669:T/TA”) in the MDS sample. We set the *WT* clone (one with reference calls for “chr21:36231782:C/T”, “chr4:106155669:T/TA”) as the baseline (“reference”) for the ploidy calling. By normalizing all other cells to the WT clone, we obtained the ploidy per amplicon. As a result, the heat maps (Figure 2C,D) revealed a loss of amplicons on chromosome 21 within the *RUNX1* region in the C3 (*TET2^N191Kfs*4^/RUNX1^R201Q_homozigous^* double-mutated) clone. The alternative visualization using a profile plot specific to the analyzed clone confirmed an LOH event. This confirmation is due to the observation that two consecutive amplicons have the same ploidy, providing additional confidence in the result (Figure 2E).

The clonal trend analysis, illustrated in Figure 3A,B, showed that therapy for MDS was able to hit the dominant clone with the single-hit *TET2^N191Kfs*4^* (C1), while the other two minor clones carrying co-occurring mutations of *TET2^N191Kfs*4^* and *RUNX1^R201Q^* (C2–C3) expanded and showed branched evolution with the acquisition of *FLT3^D835H^*, *KIT^D816H^*, *KIT^D816V^* and *NRAS^G12A^* mutations. Therefore, during disease transformation from MDS to AML, we observed a significant alteration in clonal architecture characterized by the replacement of the dominant *TET2* single-mutated clone (C1) with triple-mutated clones that evolved *via* the emergence of the *FLT3-TKD* mutation in the pre-existent double-mutated clones *TET2^N191Kfs*4^/RUNX1^R201Q^_HET* and *TET2^N191Kfs*4^/RUNX1^R201Q^_HOM* (C2–C3). In addition, AML development was driven by the branching evolution of the *TET2^N191Kfs*4^/RUNX1^R201Q^_HOM* clone through the acquisition of two independent mutations in the *NRAS* and *KIT* genes, with the emergence of two independent new subclones (C6 and C7, respectively). The analysis of AML progression samples on gilteritinib treatment demonstrated that gilteritinib therapy was able to reduce the major clones in the AML diagnosis sample (C2, C3, C4, and C5) characterized by *TET2* and *RUNX1* mutations, with or without the *FLT3-TKD* mutation. After this initial partial response, we observed, in the second f-up sample (relapse post-TKI therapy), an expansion of all mutated clones except for those carrying *FLT3-TKD* mutations, which persisted at low levels (Figure 3A,B). In particular, the clone sizes at several time points during therapy illustrated that the *KIT* and *NRAS* mutant populations detected at AML onset remained initially stable and then expanded after gilteritinib exposure. By analyzing the VAFs of mutations at different time points, we could assess the clonal evolution over the course of the disease and treatment (Figure 3C). It should be noted that in this case, the *NRAS* mutation arose in an *FLT3-WT* clone, unlike cases with *FLT3-ITD* mutations that are usually characterized by *FLT3/NRAS* double-mutated clones. This result demonstrates that pre-existing clone populations carrying *KIT* and *NRAS* mutations can already be detected at low levels at diagnosis and can be positively selected for gilteritinib, suggesting a proliferative advantage of these cells in the presence of the drug.

In the case of Pt #2, scDNAseq was performed at baseline, during first-line therapy with chemotherapy in combination with midostaurin, and at CR. A total of 23,826 cells were sequenced across three time points (10,477, 1140, and 12,209 for each time point). The analysis of clonal architecture of Pt #2 showed the presence of three clonal populations (Figure 4A). Among them, we detected two quadruple-mutated clones which differ by the presence of two distinct ITD mutations in the *FLT3* gene, and one triple-mutated clone without the *FLT3* mutation: *DNMT3A^R882C^ TET2^N1584Kfs*6^ NPM1^W288Cfs*12^ FLT3-ITD^3nt^* (C1), *DNMT3A^R882C^ TET2^N1584Kfs*6^ NPM1^W288Cfs*12^* (C2), and *DNMT3A^R882C^ TET2^N1584Kfs*6^ NPM1^W288Cfs*12^ FLT3-ITD^12nt^* (C3) (Figure 4B). Detailed description of clonal and sub-clonal populations is summarized in Table 3 and Appendix A. Analysis of clonal evolution patterns demonstrated that treatment with midostaurin in combination with intensive chemotherapy was able to induce CR by eradicating all the mutated clonal populations. Indeed, in this case, Pt #2 showed the co-presence of three clonal populations at diagnosis that were all suppressed by therapy (Figure 4C–E).

## 3. Discussion

AML is a highly heterogeneous disease, and over the last few years, intensive efforts have been devoted to the identification of recurrent genetic alterations that are potentially targetable with novel therapies. *FLT3* is one of the most recurrently mutated genes, and *FLT3* alterations are associated with clinically aggressive disease [23]. The introduction of TKIs allowing the selective targeting of *FLT3*-mutated leukemic cells holds promise to improve the traditionally dismal outcome of this setting of patients [24,25,26]. However, despite the high initial response rates, the development of resistance mechanisms may ultimately lead to treatment failure and disease progression, thereby representing a major hurdle [24,27,28]. In the era of high-throughput sequencing technologies, it is possible to investigate the genomic or transcriptomic alterations involved in drug resistance with unprecedented resolution, thereby unearthing novel therapeutic strategies for improving patient outcomes. Leukemic populations are known to leverage their inherent heterogeneity and high genetic instability to escape from targeted therapies *via* routes of clonal evolution that follow Darwinian rules. Therefore, many recent studies have focused on deciphering the role of clonal evolution in therapy resistance and disease progression and on identifying recurrent trajectories and key lesions capable of conferring selective growth advantage and increasing fitness under a given therapy. Such lesions may involve changes in the drug target itself, e.g., *FLT3* (“on-target” resistance), as well as in other genes not directly related to FLT3, yet capable to confer a survival advantage (“off-target” resistance) [29]. Although bulk sequencing has enabled crucial advances in the characterization of AML biology, it cannot provide information on clonal architecture, making it impossible to follow clone dynamics. This can now be overcome by new single-cell sequencing technologies that provide much more detailed snapshots of the intratumoral genetic heterogeneity of AML and its dynamic evolution under therapy. To date, several studies have focused on the comparison between diagnosis and relapse, showing that disease progression can follow either linear or branching evolution patterns under the pressure of treatment. In particular, many published studies have focused on *FLT3-ITD*, which is the most frequent alteration identified in AML, whereas little is known about the dynamics of *FLT3-TKD*. In this study, we used Tapestri technology to perform a longitudinal analysis on serial samples from two *FLT3*-mutated AML patients who were treated with TKIs, with opposite clinical outcomes. This pilot study demonstrated some key advantages and disadvantages of scDNAseq for the dissection and monitoring of AML genetic complexity and highlighted some interesting patterns of response and resistance to TKIs. In Pt #1, AML progression from MDS was driven by the branching evolution of two double-hit mutated clones carrying *TET2* together with *RUNX1* hetero- or homozygous mutations. The acquisition of a *FLT3-TKD* mutation (D835H) by C4 and C5 provides a potential therapeutic target. However, the high resolution of scDNAseq also allowed the identification of additional parallel minor subclones characterized by the acquisition of *NRAS* and *KIT* mutations. Therefore, unlike what was described by McMahon et al. in the *FLT3-ITD* setting [20], in our case, the *NRAS* and *KIT* mutations occurred in an *FLT3-WT* clone and were detectable at very low levels before TKI treatment. It is notable that the two clones carrying the *FLT3* mutation were not suppressed by gilteritinib treatment. It is also notable that resistance to gilteritinib was accompanied (and most likely driven) by the expansion at relapse of pre-existing subclones harboring *NRAS* and *KIT* (but not *FLT3*) mutations. This supports the hypothesis that in the presence of (certain) additional leukemic drivers, targeting FLT3 alone may not be sufficient and underlines the importance of an accurate and sensitive assessment of intratumoral heterogeneity at the clone level to inform the selection of the treatment strategy. Our case suggests that gilteritinib efficacy could be impaired by the survival and/or proliferation of leukemic cells with *RAS* and/or *KIT* mutations. In the future, sensitive screening for mutations known to be associated with resistance before and during treatment will be instrumental in predicting gilteritinib efficacy, enabling timely therapeutic intervention before overt relapse occurs, and devising alternative therapeutic targets. 

In the case of Pt #2, we identified two parallel clones characterized by distinct ITD insertions, co-occurring in both cases with *DNMT3A*, *TET2*, and *NPM1* mutations. It is important to note that, despite the relatively high number of sequenced cells, scDNAseq was not sensitive enough to detect the persistence of the *NPM1* mutation at first f-up, which was, in contrast, detected by the real-time RT-PCR approach routinely used for MRD purposes. 

Figure 5A,B recapitulates the key findings of our analysis, with clone composition and evolution over the course of treatment. Despite the limitations of analyzing only two patients and the use of different TKIs, which does not allow a direct comparison between these cases, our data highlight the fact that the presence of a complex clonality is associated with more aggressive disease. Therefore, a deeper characterization of disease biology before and during treatment could be crucial in predicting therapy response and resistance dynamics. However, clonal complexity does not necessarily mirror greater genomic instability; therefore, the propensity to acquire new mutations that can trigger a relapse seems to be related to the specific molecular features or clone composition of each individual patient.

Single-cell analysis is likely to play a pivotal role in the molecular characterization of patients with AML. However, despite the great potential of this technique, it is limited in that it interrogates only predefined hotspot regions of select genes included in the panel. In our case, the Tapestri AML panel we used for scDNAseq did not cover the *TET2^E843*^*, *RUNX1^S322Nfs*1^*, *ASXL1^E646Wfs*12^*, and *SRSF2^P95H^* gene mutations that were identified by the NGS Myeloid Solution panel performed for diagnostic routine on the MDS sample of Pt #1. Therefore, we were unable to understand how these mutations contributed to clonal complexity or how clones harboring these mutations evolved during therapy. Another limitation of scDNAseq is its high cost, which currently limits its extensive application in routine diagnostics. However, both these drawbacks are likely to be resolved in the near future.

In conclusion, although this study is limited by the analysis of only two patients, it provides a piece of knowledge about the dynamics of *FLT3-TKD* AML, showing that the presence of alterations in *NRAS* and/or *KIT* genes could predict resistance to gilteritinib. Future studies of longitudinal samples from larger cohorts of patients are warranted to paint an accurate picture of pre-existing and acquired genetic lesions impacting the efficacy of various TKIs in AML and pave the way for the integration of scDNAseq into routine diagnostics. 

## 4. Materials and Methods

### 4.1. Patients, Samples, and Cell Preparation

All patients included in the study provided written informed consent for biological studies and were treated following the local recommendations for clinical trials or routine clinical practice at the Seràgnoli Institute, IRCCS Azienda Ospedaliero-Universitaria of Bologna, Italy. The study was conducted in accordance with the Declaration of Helsinki and approved by the local Ethics Committee (CE-AVEC) of Bologna (protocol code 112/2014/U/Tess of Policlinico Sant’Orsola-Malpighi). Samples were obtained from two patients with AML carrying *FLT3* mutations at the time of diagnosis. Mononuclear cells (MNCs) from diagnostic BM aspirates were isolated by density gradient centrifugation on Ficoll-Paque Plus (Amersham Biosciences, Piscataway, NJ, USA). Selected viable cells were cryopreserved in liquid nitrogen for subsequent single-cell analysis. For diagnostic routine molecular analysis, MNCs were lysed in guanidine-thiocyanate-containing lysis buffer (RLT, Qiagen, Ltd., Milan, Italy) and stored at −20 °C until use.

### 4.2. Bulk Next-Generation DNA Sequencing (NGS) 

Genomic DNA was extracted from MNCs that were sampled from the same blood draws as the ones for single-cell DNA sequencing with Maxwell® CSC Whole Blood DNA KIT (Promega, Madison, WI, USA) according to the manufacturer’s instructions. Genomic DNA was stored at −20 °C until use. Libraries were generated using the Myeloid Solution panel (SOPHiA Genetcs; Arrow Diagnostics). The Myeloid Solution panel covers the complete coding sequence of the 30 most clinically relevant genes associated with myeloid malignancies (https://www.sophiagenetics.com/, assessed on 30 July 2024). NGS was performed on a MiSeq™ System (Illumina Inc., San Diego, CA, USA), and bioinformatic analyses were conducted using the SOPHiA DDM™ software (Sophia Genetics, Lausanne, CH, Switzerland).

### 4.3. Single-Cell DNA Sequencing (scDNAseq)

ScDNAseq was performed on the Tapestri® platform (Mission Bio, Inc., San Francisco, CA, USA), using a catalog panel (AML panel, 127 amplicons with a size ranging between 175 and 275 bp) to assess recurrent hotspot mutations in *ASXL1*, *DNMT3A*, *EZH2*, *FLT3*, *GATA2*, *IDH1*, *IDH2*, *JAK2*, *KIT*, *KRAS*, *NPM1*, *NRAS*, *PTPN11*, *RUNX1*, *SF3B1*, *SRSF2*, *TET2*, *TP53*, *U2AF1*, and *WT1* genes, according to the manufacturer’s instructions and as previously reported [30]. Sample indexes and Illumina adaptor sequences were then added by a 10-cycle PCR reaction, and the amplified sample was purified a second time with Ampure XP beads (Beckman Coulter, Inc., Brea, CA, USA). Libraries were quantified using Qubit (Thermo Fisher Scientific Inc., Waltham, MA, USA) and analyzed on a DNA 1000 assay chip with a TapeStation 4200 (Agilent Technologies Inc., Santa Clara, CA, USA). Lastly, a single sequencing run was performed for each library on an Illumina NextSeq^TM^ 550 system (Illumina, Inc., San Diego, CA, USA) with a 150 bp paired-end setting.

### 4.4. Data Processing and Variant Filtering

Sequenced data were processed using Mission Bio’s Tapestri Pipeline. Data analyses were performed using Mission Bio’s portal (https://portal.missionbio.com/, assessed on 30 July 2024) and Mission Bio’s Tapestri Insight software package (Version 2.2). Initial steps for filtering low-quality cells or genotypes were carried out with default parameters, which included removing genotypes in cells with quality lower than 0, read depth lower than 10, alternative allele frequency lower than 20, variants genotyped in less than 50% of cells, cells with less than 50% of genotypes present, and variants mutated in less than 1% of cells. Subclones were identified using Tapestri Insights 2.2 using the selected variants. Major clones were defined by the subclones that showed genotype in at least 1% of the cells. Allele dropouts (ADO) were identified when one or both alleles were not present and were excluded from the analysis.

### 4.5. Variant Calling

For subclone identification, we selected known clinical variants that are annotated as “Pathogenic” or “Likely Pathogenic” in the ClinVar database (https://www.ncbi.nlm.nih.gov/clinvar/, assessed on 30 July 2024). The DANN score system was used as the first filter step to assess the pathogenicity of a given variant (i.e., cut-off DANN score = 1 to filter out all non-pathogenic variants). From the good-quality variants, we selected coding variants for downstream analysis. Pathogenic variants were labeled as such based on (1) the coding impact of mutation (e.g., missense, nonsense, and frameshift) or (2) the pathogenicity interpretation in a clinical database (e.g., CLINVAR). After filtering, we focused only on the detected pathogenic variants by excluding common variants in the human population (freq. > 1%).

### 4.6. Clonal Architecture Analysis

Clonal architectures were initially determined by genotype clustering analysis, including zygosity information, using the Tapestri Insight software (version 2.2, Mission Bio, Inc., San Francisco, CA, USA). We included somatic coding (nonsynonymous) variants and cells with complete genotypes, whereas subclones with a missing genotype and with <1% of cells were removed. The variant metrics (VAF, genotype quality, and read depth) of each clone were inspected to identify small clones that were likely the result of allele dropout (ADO). Small clones below the ADO sample rate (cut-off < 1%) genotyped as WT or homozygous for a given variant, but with decreased quality and read depth, were considered false positives.

### 4.7. Copy Number Analysis (CNA)

Copy number analysis was performed using the Mission Bio Mosaic package version 2.3. The per-amplicon read counts were normalized to correct for systemic artifacts, first within the same cell across different amplicons by the mean read depth, and then within the same amplicon across different cells by the median read depth. Normalized reads were converted to copy number estimates by assuming that the WT clone was diploid, and all other cells normalized to read counts were scaled to the counts of this clone to derive copy number estimates. This step was performed using the compute ploidy method in the Mosaic package. Note that the median read depth across different cells is only considered good-quality cells, which are defined as those with at least 1/10 of the number of reads as that of the cell with the 10th rank in terms of the read count. Cells were clustered according to the grouping of the variants, and per-gene CNAs were called for every clone only if *n* > 2 amplicons of a given gene showed an amplified (CN > 2.5) or deleted (CN < 1.5) copy number signal.

## 5. Conclusions

Targeted therapy with TKIs is improving outcomes for patients with FLT3-mutated AML; as of 2022, the high allelic ratio is no longer considered a high-risk feature [9]. Nevertheless, resistance to FLT3 inhibitors remains a challenge. AML is known to be highly heterogeneous, with multiple subclones coexisting and evolving during treatment. Here, we show evidence that greater clonal complexity may be associated with a more aggressive disease. We also show evidence that in the presence of multiple clones harboring key leukemic drivers like *KIT* or *RAS*, FLT3 targeting alone with gilteritinib is not sufficient to control the disease. The cases described herein show that, despite the current limitations related to costs and gene panel composition, scDNAseq will become instrumental for sensitive and accurate characterization of AML disease heterogeneity and clonal architecture at diagnosis and at key therapeutic decision timepoints.

## Figures and Tables

**Figure 1 ijms-25-10002-f001:**
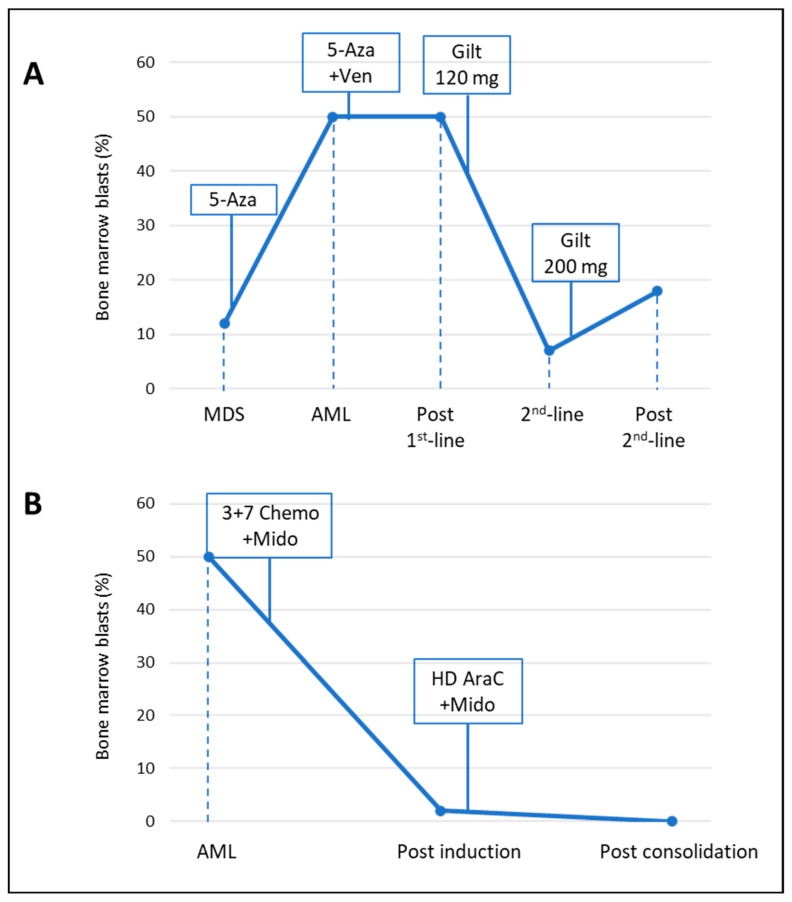
Bone marrow blasts (%) and treatments received at different time points for patient 1 (**A**) and patient 2 (**B**). The blue dotted lines indicate the time point relative to the percentage of blasts, and the boxes indicate the therapy administered. Abbreviations: 5-AZA = Azacitidine; Ven = venetoclax; Gilt = gilteritinib; Chemo = Chemotherapy; Mido = midostaurin; HD AraC = high-dose cytarabine; MDS = myelodysplastic syndrome; AML = acute myeloid leukemia.

**Figure 2 ijms-25-10002-f002:**
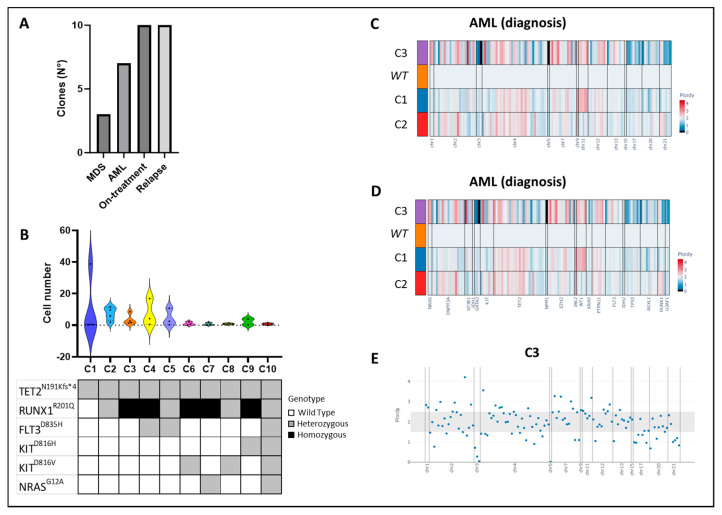
Clonal and sub-clonal populations were identified by single-cell DNA sequencing in Pt #1 at different time points. (**A**) Number of identified mutated clones in each sample analyzed at the diagnosis of MDS, AML progression, treatment, and relapse. (**B**) Distribution of clonal populations with a given genotype identified across different time points (top panel). The heat map indicates the mutated genes and zygosity for each clone (bottom panel); gray boxes represent heterozygous mutations, black boxes represent homozygous mutations, and wild-type (WT) genotypes are shown with white boxes. (**C**) Heat map of the AML diagnosis sample shows chromosome ploidy (columns) of WT, C1, C2, and C3 clonal populations (rows). (**D**) Heat map of the AML diagnosis sample shows gene ploidy (columns) of the WT, C1, C2, and C3 clonal populations (rows). (**E**) Profile plot of C3 in the AML diagnosis sample shows ploidy (y-axis) for each amplicon ordered per chromosome (x-axis). Each dot is the median ploidy per amplicon. Abbreviations: C1 = *TET2^N191Kfs*4^* single-mutated clone; C2 = *TET2^N191Kfs*4^/RUNX1^R201Q^_HET* double-mutated clone; C3 = *TET2^N191Kfs*4^/RUNX1^R201Q^_HOM* double-mutated clone.

**Figure 3 ijms-25-10002-f003:**
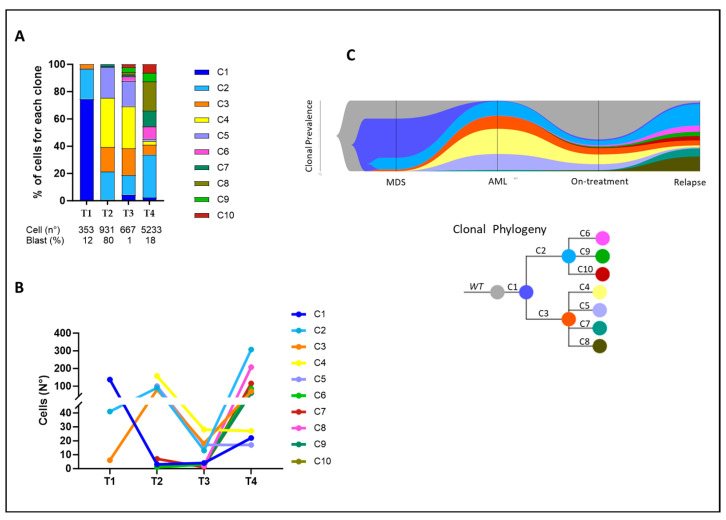
Clonal architecture and evolution shown by single-cell DNA sequencing. (**A**) Clonal and sub-clonal populations identified in Pt #1. Each column represents a specific time point, and the different colours represent the unique clonal populations identified. The total number of cells sequenced and the blast percentage for each sample are listed under the bar graph. (**B**) Patterns of clonal evolution during disease progression. Number of cells with the unique assigned genotype alongside the different time points analyzed. Legend: T1 = MDS diagnosis, T2 = AML diagnosis, T3 = during gilteritinib treatment, T4 = Relapse. (**C**) Fishplot visualizing patterns of clonal evolution according to variant allele frequencies at MDS diagnosis, AML progression, during gilteritinib treatment, and at relapse. Abbreviations: MDS = myelodysplastic syndrome; AML = acute myeloid leukemia; WT = wild type.

**Figure 4 ijms-25-10002-f004:**
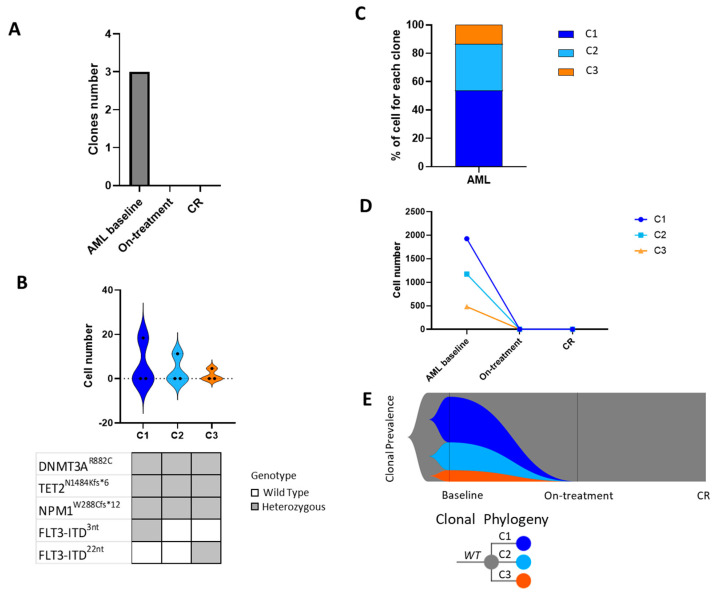
Clonal and sub-clonal populations identified by single-cell DNA sequencing in Pt #2 at different time points. (**A**) Number of mutated clones identified in each sample analyzed: at the diagnosis of AML, on treatment, and at complete remission (CR). (**B**) Distribution of clonal populations with a given genotype identified across the different time points (top panel). The heat map shows the mutated genes and zygosity of each clone (bottom panel). Gray boxes represent heterozygous mutations, whereas wild-type (WT) genotypes are shown with white boxes. Legend: T1 = AML diagnosis, T2 = intensive chemotherapy plus midostaurin, T3 = complete remission. (**C**) Clonal and sub-clonal populations identified in Pt #2. The bar graph refers to the baseline sample, and the different colors represent the unique clonal populations identified. (**D**) Patterns of clonal evolution during disease progression. Number of cells with the unique assigned genotype at the different analyzed time points. (**E**) Fish plot visualizing patterns of clonal evolution according to variant allele frequencies at AML diagnosis, during first-line treatment, and at complete remission (CR). Abbreviations: CR = complete remission; WT = wild type.

**Figure 5 ijms-25-10002-f005:**
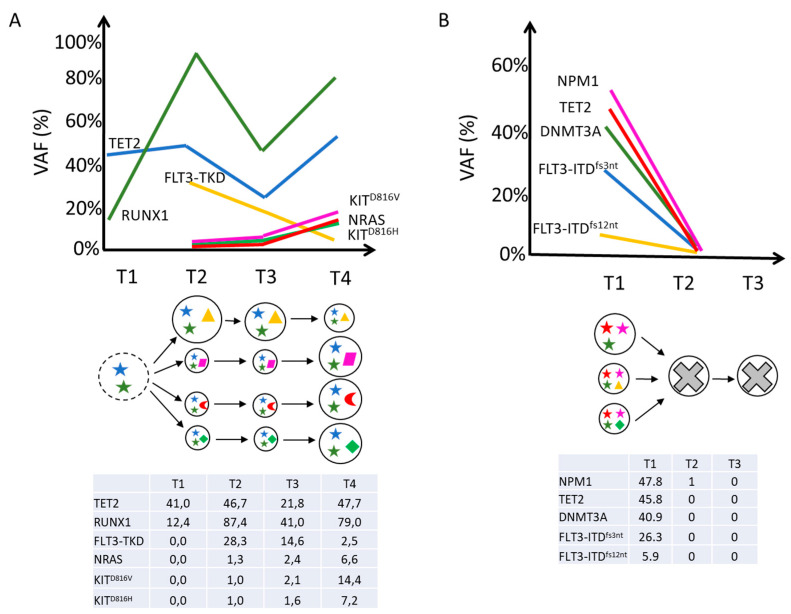
Representation of clonal evolution in each patient. Each line corresponds to an individual mutation and illustrates the presence of the mutation at each analyzed time point. Each circle corresponds to an individual cell clone defined by an identical set of mutations. Somatic mutations are represented by different symbols and colors. Cells with the “X” symbols represent wild-type cells without somatic mutations detectable by scDNAseq. The table indicates the variant allele frequencies (VAF, %) of the mutated genes in each analyzed sample (bottom panel). (**A**) Genetic evolution of Pt #1 with stable mutations and acquired mutations. Legend: T1 = myelodysplastic syndrome (MDS) diagnosis; T2 = acute myeloid leukemia (AML) diagnosis; T3 = on-treatment; T4 = relapse. (**B**) Genetic evolution of Pt #2 with lost mutations. Legend: T1 = acute myeloid leukemia (AML) diagnosis; T2 = on-treatment; T3 = complete remission. Abbreviations: VAF, variant allele frequency.

**Table 1 ijms-25-10002-t001:** Patient characteristics at study entry (AML diagnosis).

	Pt 1	Pt 2
Gender	Male	Female
Age at diagnosis (y)	82	50
AML type	Secondary to MDS	De novo
WBC at diagnosis (10^3^ × µL)	111.370	30.000
Peripheral blasts (%)	69	45
Bone marrow blasts (%)	50	50
ELN Risk category	Adverse	Intermediate
*FLT3* mutational status	TKD positive (D835H)	ITD positive
Additional mutations	*ASXL1*, *SRSF2*, *TET2*, *RUNX1*	*NPM1*
Cytogenetics	NK (46, XY)	NK (46, XX)
First-line therapy	azacitidine plus venetoclax	Intensive chemotherapy plus midostaurin
Second line therapy	gilteritinib	N.A.
Consolidation therapy	N.A.	High-dose cytarabine plus midostaurin

Abbreviations: AML, acute myeloid leukemia; MDS, myelodysplastic syndrome; WBC, white blood cell; ELN, European Leukemia Net; ITD, internal tandem duplication; TKD, tyrosine kinase domain; NK, normal karyotype; N.A., not available.

**Table 2 ijms-25-10002-t002:** Description of clonal populations and their percentages at different time points on Pt #1.

Clone	Pathogenic Variants	Time Point	Proportion (%)
C1	*TET2^N191Kfs*4^*	T1	38.1
		T2	0.32
		T3	0.6
		T4	0.42
C2	*TET2^N191Kfs*4^ RUNX1^R201Q^_HET*	T1	11.61
		T2	9.67
		T3	1.94
		T4	5.87
C3	*TET2^N191Kfs*4^ RUNX1^R201Q^ _HOM*	T1	1.7
		T2	8.59
		T3	2.68
		T4	1.34
C4	*TET2^N191Kfs*4^*; *RUNX1^R201Q^ _HOM*; *FLT3^D835H^*	T1	0
		T2	16.86
		T3	4.17
		T4	0.52
C5	*TET2^N191Kfs*4^ RUNX1^R201Q^_HET*; *FLT3^D835H^*	T1	0
		T2	10.74
		T3	2.53
		T4	0.32
C6	*TET2^N191Kfs*4^ RUNX1^R201Q^ _HOM*; *NRAS^G12A^*	T1	0
		T2	0.11
		T3	0.45
		T4	1.62
C7	*TET2^N191Kfs*4^ RUNX1^R201Q^_HOM*; *KIT^D816H^*	T1	0
		T2	0.75
		T3	0.15
		T4	2.2
C8	*TET2^N191Kfs*4^ RUNX1^R201Q^ _HOM*; *KIT^D816V^*	T1	0
		T2	0
		T3	0.3
		T4	3.96
C9	*TET2^N191Kfs*4^ RUNX1^R201Q^_HET*; *KIT^D816H^*	T1	0
		T2	0
		T3	0.45
		T4	1.17
C10	*TET2^N191Kfs*4^ RUNX1^R201Q^_HET*; *NRAS^G12A^*	T1	0
		T2	0
		T3	0.3
		T4	1.17

T1 = myelodysplastic syndrome (MDS) diagnosis; T2 = acute myeloid leukemia (AML) diagnosis; T3 = on-treatment; T4 = relapse. Abbreviations: HET = heterozygous mutation; HOM = homozygous mutation.

**Table 3 ijms-25-10002-t003:** Description of clonal populations and their percentages at different time points on Pt #2.

Clone	Pathogenic Variants	Time Point	Proportion (%)
C1	*DNMT3A^R882C^ TET2^N1584Kfs*6^ NPM1^W288Cfs*12^ FLT3-ITD^3nt^*	T1	18.38
		T2	0
		T3	0
C2	*DNMT3A^R882C^ TET2^N1584Kfs*6^ NPM1^W288Cfs*12^*	T1	11.21
		T2	0
		T3	0
C3	*DNMT3A^R882C^ TET2^N1584Kfs*6^ NPM1^W288Cfs*12^ FLT3-ITD^12nt^*	T1	4.58
		T2	0
		T3	0

Legend: T1 = acute myeloid leukemia (AML) diagnosis; T2 = on-treatment; T3 = complete remission.

## Data Availability

Data and materials described in this manuscript, including all relevant raw data, will be freely available from the corresponding author.

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
