# Peer review of "Tracking Response and Resistance in Acute Myeloid Leukemia through Single-Cell DNA Sequencing Helps Uncover New Therapeutic Targets"

_ijms, 2024, doi:10.3390/ijms251810002_

Round 1

Reviewer 1 Report

Comments and Suggestions for Authors

The authors describe a proof-of-concept of tracking response profiles in patients with acute myeloid leukemia using modern next generation sequencing technology at single cell DNA level.

This represents an effort in moving science towards precision medicine and the implementation of such technologies only acquires more interest if many articles envision the use of such technologies.

My comments are below:

-The Abstract is a bit unclear to the reader. The reader expects many samples subjected to scDNA seq. Please revise the abstract presenting the number of patients studied and the actual results which were found.

-What was the cytogenetics of the cases at diagnosis? Please include it in Table 1.

-Did the authors detect any other chromosomal lesions using scDNA seq? Considering the sensitivity, other lesions could be detected.

-In the Methods section, the panel of genes (scDNA seq) does not include TET2  gene? Is this true? If true, the tracking representation will be largely skewed considering TET2 mutations being present in the original specimen.

-How many small clones (cut-off less than 1%) were eliminated according to ADO? i.e. were false positive?

Minor comments:

-Gene names should be in Italic font

-Spaces along the text should be checked (mainly abstract section). Words are often attached each other.

Comments on the Quality of English Language

English grammar should be checked.

Author Response

1-The Abstract is a bit unclear to the reader. The reader expects many samples subjected to scDNA seq. Please revise the abstract presenting the number of patients studied and the actual results which were found.

Response 1: We thank the Reviewer for the remark, we have revised the abstract as suggested.

2-What was the cytogenetics of the cases at diagnosis? Please include it in Table 1.

Response 2: We added the cytogenetics in Table 1.

3-Did the authors detect any other chromosomal lesions using scDNA seq? Considering the sensitivity, other lesions could be detected.

Response 3: No, we did not detect additional chromosomal lesion. AML panel used in the study is not ideal for chromosomal analysis, it could only detect copy number variation of genes included in the panel.

4-In the Methods section, the panel of genes (scDNA seq) does not include TET2 gene? Is this true? If true, the tracking representation will be largely skewed considering TET2 mutations being present in the original specimen.

Response 4: Yes, the panel indeed includes the TET2 gene (https://missionbio.com/products/panels/acute-myeloid-leukemia/). I apologize for the oversight; we have modified the list of genes in the Methods section.

5-How many small clones (cut-off less than 1%) were eliminated according to ADO? i.e. were false positive?

Response 5: We did not eliminate small clones, i.e. for pt#1 the smallest clone was at 0.3 %. The ADO occurs when a sample is genotyped and one or both alleles are not present, therefore mutation percentage is around 100% and is considered false positive because of the loss of wild type allele. In this case the CNV analysis allow to distinguish ADO from CNV.

6-Minor comments:

-Gene names should be in Italic font

-Spaces along the text should be checked (mainly abstract section). Words are often attached each other.

Response 6: We thank the Reviewer, we have now revised the manuscript according to his/her feedback.

Reviewer 2 Report

Comments and Suggestions for Authors

I have the following concerns:

1. Since the tables describing the dynamics of the AML mutations are complicated, please include a novel table stating the MAIN findings of your study.

2. Please include the restrictions of your study to the discussion (only 2 patients studied etc).

3. Is the method used to identify the dynamics and kinetics of individual mutations expensive? What is the cost for screening every patient with single cell analysis?

4. Table 1. Patient 2 characteristics. I think that high dose cytarabine plus midostaurin is not second line therapy, but a continuation of first line therapy, since this patient did not relapse from the initial intensive chemo plus midostaurin. This is not the case with patient 1, who indeed relapsed and the table is correct for patient 1. Please correct the table for patient 2. 

5. Abstract. A grammatical error. Development of TKIs has improving......please correct has improved. 

6. Please make the discussion clear, more precise and more simple.  

7. Usually the materials and methods section is after the Introduction and before the Results. 

Comments on the Quality of English Language

Some minor language editing needed. 

Author Response

  1. Since the tables describing the dynamics of the AML mutations are complicated, please include a novel table stating the MAIN findings of your study.

Response 1: We thank the Reviewer for the suggestion. We simplified the table by leaving only the percentages of clones identified for each time point. The complete tables are included as supplementary.

  1. Please include the restrictions of your study to the discussion (only 2 patients studied etc).

Response 2: We thank the Reviewer for the suggestion. We have modified the text in order to include the restriction of the study.

  1. Is the method used to identify the dynamics and kinetics of individual mutations expensive? What is the cost for screening every patient with single cell analysis?

Response 3: Yes, the method used is expensive. The cost of reagents is about 6000 € for each sample. We have included a comment on the cost of the technique in the discussion section.

  1. Table 1. Patient 2 characteristics. I think that high dose cytarabine plus midostaurin is not second line therapy, but a continuation of first line therapy, since this patient did not relapse from the initial intensive chemo plus midostaurin. This is not the case with patient 1, who indeed relapsed and the table is correct for patient 1. Please correct the table for patient 2. 

Response 4: We have changed the table, putting high dose cytarabine plus midostaurin as consolidation therapy.

  1. Abstract. A grammatical error. Development of TKIs has improving......please correct has improved. 

Response 5: We have corrected it.

  1. Please make the discussion clear, more precise and more simple.  

Response 6: As suggested by the Reviewer, we have substantially changed the discussion to make it more clear and precise.

  1. Usually the materials and methods section is after the Introduction and before the Results. 

Response 7: The various sections of the article are organized according to the template provided by the journal.

Reviewer 3 Report

Comments and Suggestions for Authors

To "highlight the importance of a deeper characterization of patient’s clonal architecture for clinical outcome prediction and for development of personalized therapy" is not providing any reason to read the article.

The only conceivable way to make the article to provide added value is to improve the discussion, and provide  1) clarity;  2) effective concepts;  3) scientific depth.

Add concept images. At least one for each important concept.

Fix spaces in the text. For example:

cells (HSCs)[1], [2].The FMS-like tyrosine kinase 3 (FLT3)gene is frequently altered in 31

AML,is a crucial player in leukemogenesisand confers a dismal progno- 32

sis.Physiologically, FLT3 is expressed by hematopoietic stem and progenitor cells and is 33

High throughputscDNAsequsing theTapestri platform was performed on mononuclear 125

cells isolated fromPt #1 at different time points:at the time of initialMDS diagnosis, at the 126

Author Response

  1. To "highlight the importance of a deeper characterization of patient’s clonal architecture for clinical outcome prediction and for development of personalized therapy" is not providing any reason to read the article.

Response 1: We thank the Reviewer for the suggestion. We have changed the abstract trying to make it more appealing.

  1. The only conceivable way to make the article to provide added value is to improve the discussion, and provide  1) clarity;  2) effective concepts;  3) scientific depth.

Response 2: As requested by the Reviewer, we have substantially changed the discussion to make it more clear and precise.

  1. Add concept images. At least one for each important concept.

Response 3: As suggested by the Reviewer, we have added 2 concept image (5A-B) that summarized our results.

  1. Fix spaces in the text.

 Response 4: We thank the Reviewer; we have now revised the manuscript to fix spaces.

Round 2

Reviewer 3 Report

Comments and Suggestions for Authors

Despite the improvements, the article still lacks focus and depth. In addition to this, the syntax used does not allow the reader to draw conclusions.

Example: "Despite the high potential of this technique to dissect the clonal landscape of 331
leukemic diseases, it is limited in that it interrogates only pre-defined hot-spot regions of 332
the targeted genes included in the panel. Therefore, in our case, the AML panel used for 333
scDNAseq does not cover the TET2E843*, RUNX1S322Nfs*1, ASXL1E646Wfs*12 and SRSF2P95H gene 334
mutations identified by the NGS Myeloid Solution panel performed in diagnostic routine 335
on MDS sample of Pt #1 (https://designer.missionbio.com/catalogpanels/AML). Therefore, 336
we were unable to characterize the undetected mutations and define their evolution."

- Make corrections to errors such as "mutations identified for each patients". The correct use is "each patient".

- Proof read the entire article.

Other,

tentative suggestion: Perhaps consider copying and adding the phrase " data highlight the fact that the presence of a complex clonality is associated 348
with more aggressive disease. " also to other parts of the article, such as the conclusions.

Comments on the Quality of English Language

Some proof reading is needed. Also, attention to the phrases that connect subsequent sentences, will help the reader understand why this article is important.

Author Response

1 -Despite the improvements, the article still lacks focus and depth. In addition to this, the syntax used does not allow the reader to draw conclusions.

Example: "Despite the high potential of this technique to dissect the clonal landscape of 331
leukemic diseases, it is limited in that it interrogates only pre-defined hot-spot regions of 332
the targeted genes included in the panel. Therefore, in our case, the AML panel used for 333
scDNAseq does not cover the TET2E843*, RUNX1S322Nfs*1, ASXL1E646Wfs*12 and SRSF2P95H gene 334
mutations identified by the NGS Myeloid Solution panel performed in diagnostic routine 335
on MDS sample of Pt #1 (https://designer.missionbio.com/catalogpanels/AML). Therefore, 336
we were unable to characterize the undetected mutations and define their evolution."

- Make corrections to errors such as "mutations identified for each patients". The correct use is "each patient".

- Proof read the entire article.

Response 1. We thank the Reviewer for the suggestions. We have now revised the entire manuscript according to his/her feedback. In the revised text, we modified the syntax to improve clarity (e.g. revised text in lines 345-351). Moreover, as suggested, we have modified the Discussion and Conclusion sections in order to provide more focus and depth.

2 -Other, tentative suggestion: Perhaps consider copying and adding the phrase " data highlight the fact that the presence of a complex clonality is associated 348
with more aggressive disease. " also to other parts of the article, such as the conclusions.

Response 2. As suggested by the Reviewer, we have added the sentence in the conclusion section (revised text on page 14).

3 -Some proof reading is needed. Also, attention to the phrases that connect subsequent sentences, will help the reader understand why this article is important.

Response 3. As suggested by the Reviewer, we performed proof reading of the entire manuscript, all corrections were made by track change.
